# Influence of PSD Estimation Parameters on Fatigue Life Prediction in Spectral Method

**DOI:** 10.3390/ma16031007

**Published:** 2023-01-21

**Authors:** Adam Kaľavský, Adam Niesłony, Róbert Huňady

**Affiliations:** 1Department of Applied Mechanics and Mechanical Engineering, Faculty of Mechanical Engineering, Technical University of Košice, 04 200 Košice, Slovakia; 2Department of Mechanics and Machine Design, Faculty of Mechanical Engineering, Opole University of Technology, 45-271 Opole, Poland

**Keywords:** power spectral density, Welch method, spectral fatigue, probability density function, sensitivity analysis

## Abstract

The paper analyzes in detail the influence of the Welch method on the estimation of fatigue life in the frequency domain. The aim is to examine how the setting of parameters, such as window size, weighting and overlap, affects the calculation of power spectral density (PSD) and thus lifetime. The paper uses knowledge from the theory of signal processing, statistics and fatigue life estimation in the frequency domain. In order to demonstrate the general validity of the conclusions, four different reference PSD spectra are generated and examined. The spectra are converted to time histories using an inverse fast Fourier transform and then used as input (measured) data for the Welch method. The PSD thus obtained are subsequently subjected to lifetime estimation using the Dirlik and Tovo–Benasciutti models. Based on the obtained results, it was found that the best configuration of the parameters of the Welch method for fatigue calculations is the selection of the smallest possible time window (maximum 60% of the observation length) and the largest possible overlap (minimum 70% of the window size).

## 1. Introduction

Digital signal processing is one of the most powerful technologies that has shaped science and engineering for the last five decades. The interfacing of measurement instrumentation and computers has brought significant changes in procedures such as data acquisition, digitization, storage, processing and evaluation [1]. Many methods, algorithms and techniques have been developed to manipulate digital signals and data, and which have led to the simplification of many processes and analyses commonly used in research and development. Many of them are so well implemented that few scientists are involved in verifying them. One of these methods is the algorithm for calculating the power spectral density (PSD), which has a wide application in various fields. In structural mechanics, PSD is used to identify modal parameters [2,3], to determine vibration responses [4,5], to detect damage [6,7] or to estimate fatigue life [8,9,10].

In practice, many mechanical structures and components (such as vehicles, wind turbines, PCBs, etc.) are exposed to random vibration during operation. In these cases, the assessment of fatigue life using frequency domain methods is advantageous because it provides a direct connection between the power spectral density and the damage intensity and significantly speeds up calculations compared to time domain analysis [10,11]. However, the frequency methods are effective when the PSD is properly determined and describes the random load sufficiently to determine the fatigue life [12].

If the time-varying load or response is in the nature of a stationary random Gaussian process, the power spectral density can be used to described it in the frequency domain [1]. The best-known methods to obtain PSD numerically are parametric methods, non-parametric methods and subspace methods.

In parametric methods, the signal is assumed to be an output of a linear system driven by white noise, i.e., parametric methods estimate the PSD by initially estimating the parameters of the system assumed to generate the signal. Autoregressive (AR), moving-average (MA), or autoregressive moving average (ARMA) time-series models can be used to describe the system [13,14]. Therefore, parametric methods also are known as model-based methods. To estimate the PSD of a time series, it is necessary to obtain the model parameters, e.g., using the Yule–Walker or Burg’s method. The parametric methods are used mainly because they produce PSD with a finer frequency resolution than traditional periodogram techniques. However, they are sensitive to modelling errors, signal-to-noise ratio and, in most cases, have a large computational load.

The non-parametric methods refer to the methods of estimating the spectral density of a random signal without pre-parameter modelling and are therefore robust and much less sensitive to noise. They are based on the relationship between the PSD and the autocorrelation function, which are a Fourier pair [14]. The PSD is estimated directly from the signal itself using the Fourier transform. One of the simplest ways to estimate PSD is to find the discrete-time Fourier transform of the samples of the signal and appropriately scale the magnitude squared of the result. This estimate method is called the periodogram. The Fourier transform can be performed using the fast Fourier transform (FFT) algorithm that is computationally efficient and far more accurate than other transforms. The disadvantage of fast Fourier transform is the spectral leakage that occurs when the signal being measured is not periodic in the sample interval. This effect cannot be entirely eliminated; however, it has been proven that this effect can be reduced by using weighting windows, e.g., Hanning, Hamming and others [15,16]. The periodogram has many variants, such as Bartlett’s method, Welch’s method, multitaper method and more. Presently, the Welch method [17] is widely used due to its efficiency and scalability.

A third type of method used to estimate PSD are subspace methods. They are based on eigen analysis or singular value decomposition of the autocorrelation matrix. The subspace methods are most suitable for line spectra and are effectively used to detect sinusoids or to reduce noise in a signal. However, they produce only the so-called pseudospectrum. These methods include, among others, the multiple signal classification (MUSIC) method and Pisarenko harmonic decomposition method.

One way to estimate the life under variable amplitude or random loading in the time domain is by the rainflow counting algorithm, which allows to determine load cycles (amplitudes and mean values) from a random signal, first introduced by Matsuishi and Endo [18]. Downing and Socie [19] created one of the more widely referenced and utilized rainflow cycle-counting algorithms in 1982, which is included in ASTM E 1049-85. After obtaining the cycles’ amplitude–mean distribution, the accumulation of damage is carried out according to the hypothesis of linear damage accumulation, known as the Miner rule [20]. Knowing the value of the fatigue damage, it is possible to determine the expected time until the crack initiation. This method is well described mainly for uniaxial loading. At present, researchers are focusing more on a comprehensive solution to fatigue processes, e.g., under multiaxial loading [21,22] or thermo-mechanical loading [23,24]. Significant studies have also been published on the topic of very high cycle fatigue [25,26].

Although the rainflow method is considered one of the most accurate for estimating fatigue life, several methods have been derived for estimating high-cycle fatigue in the frequency domain [27,28]. They try to obtain a cycle distribution according to rainflow counting directly from the PSD parameters in the frequency domain only. Some of them are specific and provide good estimation only for a certain type of signal (narrowband or broadband). Nevertheless, there are models which provide particularly satisfactory results in both narrowband and broadband signals. These include the Dirlik and Tovo–Benasciutti models [29].

The Dirlik method [30], devised in 1985, uses an empirical expression for calculating the probability density function, which was obtained by extensive numerical simulations using the Monte Carlo technique. It approximates the cycle–amplitude distribution by using a combination of one exponential and two Rayleigh probability densities. Benasciutti and Tovo [29,31] proposed an approach where fatigue life is calculated as a linear combination of top and down fatigue damage intensity limits.

The paper focuses on the correct estimation of the power spectral density in the sense of fatigue calculations. In the literature, it can be seen that the Welch method is the most common and even the only method used to determine the PSD for this application case. For this reason, only this method was chosen for deeper analysis. The Welch method requires several parameters that may affect the fatigue calculation process. One of the key parameters is the window type, which refers to the shape of the window function that is applied to each segment of the signal. Common window types include the rectangular window, the Hanning window, the Hamming window, and the Blackman window. Each window type has its own characteristics and may be more or less suitable for a given application, depending on the desired trade-off between spectral resolution and spectral leakage reduction. The window length is another important parameter, as it determines the length of each segment and therefore the frequency resolution of the resulting PSD estimate. A shorter window length will provide higher frequency resolution but may also result in a noisier estimate. A longer window length will provide a smoother estimate, but at the cost of lower frequency resolution. The overlap between successive segments is also a key parameter in the Welch method. A larger overlap will result in a smoother PSD estimate but will also require more computation time. A smaller overlap will provide a less smooth estimate but will be faster to compute. The optimal overlap will depend on the specific characteristics of the signal and the desired trade-off between smoothness and computational efficiency.

The main result of the article is to show which parameter of the Welch method affects the calculated fatigue life, also providing guidance on how to set these parameters to obtain reliable fatigue life estimation.

## 2. Theory Background

The PSD expresses the power contained in the narrow band Δω of the continuous spectrum and shows the power distribution along the frequency line. From a physical point of view, it is not possible to talk about the PSD of a discrete signal due to the discrete signal having no power. However, if the sampling is dense enough, it is possible to assume the PSD of the discrete signal proportional to the PSD of the continuous signal.

The basis of analytical calculation of PSD can be an autocorrelation function using Wiener–Chinchin relations. The power spectral density $S_{xx}(\omega )$ and the autocorrelation function $R_{xx}(\tau )$ (in reciprocal) are given by the formula [32]:(1)$$S_{xx}\left(\omega \right)=\int_{-\propto }^{+\infty }R_{xx}\left(\tau \right)e^{-\mathrm{j\omega \tau}}d\tau ,\;R_{xx}(\tau )=\frac{1}{2\pi }\int_{-\propto }^{+\infty }S_{xx}(\omega )e^{\mathrm{j\omega \tau}}d\omega ,$$
where ω is the frequency, and τ is the time lag. Both the functions are related to the direct and inverse Fourier transform. The correlation function $R_{xx}(0)$ with zero lag represents the signal power. It is known that the autocorrelation function is an even function. PSD also has this feature, and therefore, $S_{xx}(\omega )=S_{xx}(-\omega )$. The PSD computation can be accomplished in many ways. One of them is the Welch method, the theory of which is explained below.

The Welch method uses a modified version of the Bartlett method in which the segments of the series contributing to each periodogram may overlap each other. This means that the discrete time signal x(n) can be divided into data segments, each of which can be described by the formula:(2)$$x_{i}(n)=x\left((i-1)(N-O)+n\right),\;n=1\ldots N,\;i=1\ldots I$$
where I is the number of segments, N is the number of samples of each segment, and O is the overlap expressed in samples. The expression (i−1)(N−O) represents the starting point of the *i*-th sequence. In this way, the Welch method reduces the variance of the periodogram.

In addition, the Welch method allows the data segment to be weighted by a time window function before calculating the periodogram. Therefore, the Welch method is also called the weighted overlapped segment averaging method or the periodogram averaging method. The result is a modified periodogram given by:(3)$$PSD_{i}(k)=\frac{1}{NU}{\left|x_{i}(n)w(n)e^{-j\frac{2\pi }{N}kn}\right|}^{2}$$
where w(n) is the weighting window, and U is the normalization coefficient that expresses the power of the window and is given by:(4)$$U=\frac{1}{N}\sum_{n=1}^{N}w^{2}(n)$$

The Welch estimate of the power spectral density is an average of the modified periodograms of all segments:(5)$$PSD(k)=\frac{1}{I}\sum_{i=1}^{I}PSD_{i}(k)$$
where k is the frequency line number.

Because the segments usually overlap, data values at the beginning and end of the segment tapered by the window in one segment, occur away from the ends of adjacent segments. This guards against the loss of information caused by windowing.

A weighting window using a trigonometric cosine function, which has a value close to zero at the edges, is named after mathematician Julius von Hanns as the Hanning window. It is one of the most-used time windows. This window is defined as follows:(6)$$w\left(n\right)=0.5\times \left(1+\cos\left(\frac{2\pi n}{N-1}\right)\right),n=-\frac{N}{2},\ldots ,-1,0,1,\ldots ,\frac{N}{2}-1$$

The next well-known window designed by Richard W. Hamming is a modification of the Hanning window. It has an even greater effect on the dampening of the side lobes. It is defined by two coefficients α and β:(7)$$w\left(n\right)=\alpha +\beta \cdot \cos\left(\frac{2\pi n}{N-1}\right),n=-\frac{N}{2},\ldots ,-1,0,1,\ldots ,\frac{N}{2}-1$$

It most effectively suppresses the first sub-lobe when α=0.54 and β=1−α=0.46. To cancel the largest side-lobe, α=0.54, β=(1−α)/2=0.23 is recommended.

The power spectral densities obtained in this way are prepared for the calculation of parameters, such as spectral moments $m_{0},m_{1},m_{2},m_{4}$ and the asymmetry coefficient γ, which enter into the estimate of the probabilistic models used later to calculate the lifetime. The Dirlik probability density function of the rainflow-cycle amplitude [30] is given by:(8)$$p_{a,D}\left(S\right)=\frac{\left(\frac{D_{1}}{Q}\cdot e^{\frac{-S_{i}}{Q}}+\frac{D_{2}\cdot S_{i}}{R^{2}}\cdot e^{\frac{-S_{i}^{2}}{2R^{2}}}+D_{3}\cdot S_{i}\cdot e^{\frac{-S_{i}^{2}}{2}}\right)}{\sqrt{m_{0}}},$$
where $S_{i}$ represents the normalized stress amplitude, and the constants $D_{1},D_{2},D_{3},Q,R$ are calculated using the spectral moments $m_{0},m_{1},m_{2},m_{4}$ and the cycle asymmetry coefficient γ as follows:$$S_{i}=\frac{s}{\sqrt{m_{0}}},x_{m}=\frac{m_{1}}{m_{0}}\sqrt{\frac{m_{2}}{m_{4}}},\gamma =\frac{m_{2}}{\sqrt{m_{0}m_{4}}},$$
$$D_{1}=\frac{2\left(x_{m}-\gamma^{2}\right)}{1+\gamma^{2}},D_{2}=\frac{1-\gamma -D_{1}+D_{1}^{2}}{1-R},D_{3}=1-D_{1}-D_{2},$$
$$Q=\frac{1,25\left(\gamma -D_{3}-D_{2}R\right)}{D_{1}};R=\frac{\gamma -x_{m}-D_{1}^{2}}{1-\gamma -D_{1}+D_{1}^{2}}.$$

Another frequently used probabilistic model is the Tovo–Bennasciutti model, whose probability density function is [33]:(9)$$p_{a,B}\left(S\right)=bN_{B}+\left(1-b\right)R_{C}$$
where variables $N_{B},R_{C}$ and b are expressed as:$$b=\frac{\left(a_{1}-a_{2}\right)\left[1.112\left(1+a_{1}a_{2}-\left(a_{1}+a_{2}\right)\right)e^{2.11a_{2}}+\left(a_{1}-a_{2}\right)\right]}{{\left(a_{2}-1\right)}^{2}},$$
$$R_{C}=\frac{S_{i}}{\left(a_{2}^{2}m_{0}\right)e^{\frac{-S_{i}^{2}}{2\left(a_{2}^{2}m_{0}\right)}}},N_{B}=\frac{a_{2}S_{i}}{m_{0}}\cdot e^{\frac{-S_{i}^{2}}{2m_{0}}},a_{1}=\frac{m_{1}}{\sqrt{m_{0}m_{2}}},a_{2}=\gamma .$$

It can be noted that the Tovo–Benasciutti model uses a linear combination of the narrow-band $N_{B}$ and range counting $R_{C}$ results. When the probability density function $p_{a}(S)$ is known, then it is possible to express the expected damage intensity $\bar{D}$, according to the Palmgren–Miner rule (ignoring the mean value). The damage intensity estimates the damage per unit of time, and it is defined as [33]:(10)$$\bar{D}=\nu_{p}C^{-1}\int_{0}^{\infty }S^{k}p_{a}(S)dS,$$
where C is the S-N curve constant, k is the S-N slope coefficient, and $\nu_{p}$ is the expected peak occurrence frequency:(11)$$\nu_{p}=\sqrt{\frac{m_{4}}{m_{2}}}.$$

The fatigue-lifetime estimate in the frequency domain is obtained from the damage intensity $\bar{D}$:(12)$$T=\frac{1}{\bar{D}}.$$

## 3. Design of Experiment

The aim of the experiment was to investigate how window size, overlap, and weighting affect the estimation of fatigue life using the Welch method to calculate PSD. The range of the parameters and types of the weighting function used in the calculations are listed in Figure 1. 

In addition, to check the influence of the wave spectrum width on the power spectrum density estimates, 6 different PSD functions were used as waveform sources. The shapes of these PSD functions were selected on the basis of literature and standard recommendations, see Figure 2.

PSD was selected as the source of the time signals to ensure the same PSD shape for each time waveform during the simulation. Time courses were obtained by inverse Fourier transform with random phase shift. All calculations were performed in MATLAB.

### 3.1. Setting the Shape of PSD Functions

For analysis purposes, six different PSDs were created and used in fatigue life calculations. The first three of them were (user-defined) synthesized functions with constant, linear and quadratic character, respectively. They are suitable to examine the sensitivity of PSD to changes in the parameters of the Welch method. As fatigue analyses are relatively widespread in engineering practice, the authors considered it necessary to give a practical example of PSD. To generalize the findings, calculations were performed for six different types of PSD functions. Therefore, three other functions were taken from the technical standards ISO 16750-3 and EN 61373. ISO 16750-3 deals with mechanical loads of road vehicles with a focus on environmental conditions and electrical testing for electrical and electronic equipment. EN 61373 deals with railway applications, rolling stock equipment, shock and vibration tests. All of the above PSDs are shown in Figure 2. The fatigue life results calculated based on them are considered the exact reference solution. The last two PSDs, shown in Figure 2e,f, are defined in logarithmic scale, so they had to be converted to a linear scale to be usable for further calculations.

### 3.2. Time Signals

Since the Welch method requires time signals, it was necessary to convert the reference PSDs to the time domain. In engineering practice, signals that contain a large number of samples are usually used to describe the time history, so the histories are smooth and well defined. To achieve the same effect, the length of the PSD corresponding to the number of frequency lines (bins) was set equal to the next higher power of 2, and then upsampled several times. This extended the acquisition interval of the time signal, which was subsequently obtained by inverse fast Fourier transform (IFFT), see Figure 3. 

The sampling frequency was set so that the resolution of the frequency spectrum was 1 Hz in all cases. Since the PSDs were defined as one-sided, several adjustments had to be made before using the IFFT algorithm. First, a two-sided spectrum was calculated. A random number generator implemented in MATLAB was used to define the phase values. The generator uses the Gaussian distribution, so the results of the time history took the form of a Gaussian load. To avoid introducing inaccuracies into the calculations, the same set of random numbers was used for each signal. The conjugated symmetric vector thus obtained ensures that the IFFT provides a real value output. This approach also speeds up the discrete IFFT algorithm:(13)$$x\left(j\right)=\frac{1}{n}\sum_{k=1}^{n}X\left(k\right)\cdot W_{n}^{-\left(j-1\right)\left(k-1\right)},$$
where $X\left(k\right)$ is the discrete Fourier transform of sequence $x\left(j\right)$ and:$$W_{n}=e^{\frac{\left(-2\pi i\right)}{n}}.$$

The kurtosis function was used to ensure the highest accuracy of normal distribution. The time histories obtained by IFFT are considered to be the input (measured) time signals. An example of the time history corresponding to PSD A is shown in Figure 3.

### 3.3. PSD Calculation by Welch Method

The MATLAB function “PWELCH” was used to calculate the PSDs. The DOE algorithm was based on loops, which ensured the automatic change of selected parameters of Welch method in the range given in Figure 1. This allowed a quick comparison of the results obtained. Within the DOE analysis, the “Full factorial” technique was used, the design matrix of which is a combination of all inputs in all specified values. All PSDs were further processed to assess the impact of the individual parameters. First, the effect on spectral moments was analyzed. The spectral moments can be calculated as [34]:(14)$$m_{n}=\int_{0}^{\infty }f^{n}G\left(f\right)df=\sum f^{n}\cdot G\left(f\right)\cdot \delta f,$$
where $G\left(f\right)$ is the PSD function and δ is the discrete bandwidth of PSD (also known as frequency bin width).

For n=0, the spectral moment $m_{0}$ corresponds to the variance of the time history. Figure 4 shows the zero spectral moments for all PSDs with a Hamming window. The reference value corresponding to the reference PSDs is shown as a transparent grid plane.

As can be seen, in general, the Welch method provides a sufficiently accurate estimate of the variance for a window size of up to 60% of all samples regardless of the character of the PSD. When larger segments are used, accuracy decreases exponentially. The plots in Figure 4 also show that increasing the overlap improves the estimate, with the highest accuracy being obtained with an overlap above 50% of the window size. Optimal adjustment of the window size and overlap reduces the variance of the periodograms. This effect can be also seen in the PSD plots shown in Figure 5, where the blue line represents the reference PSD. 

The PSD of window size 20%, overlap 90% and Hamming window is shown by a red line. It is clear from the figure that the PSD calculated for these Welch method parameters shows a very good match with the reference PSD, in contrast to the PSD whose window size was 85% and the overlap was 25%. This PSD function (shown in yellow) shows significant local variations in value compared to the periodogram method, i.e. line smoothness is lost, which is undesirable when calculating the lifetime by spectral methods.

To confirm the general validity of the above findings, the same analysis was performed for the PSDs obtained using the Hanning window. The procedure was identical to the previous case with Hamming. The dependence of the zero spectral moments on the size of the window and the overlap is shown in Figure 6. A comparison of the PSDs obtained with different settings of the Welch method parameters and the reference PSD functions is shown in Figure 7. Based on the results, the same conclusions can be drawn. 

### 3.4. Fatigue Life Calculation

The fatigue life calculation was performed for two probabilistic models—Dirlik and Tovo–Benasciutti. In both models, the S-N curve of a carbon steel given in [35] was used to calculate the lifetime. For simplicity, an idealized S-N curve constantly decreasing to zero amplitude was considered (Figure 8). It was obtained by nonlinear fitting according to the Basquin’s equation that describes the relation between stress amplitude and fatigue life:(15)$$\sigma_{a}={\sigma^{\prime }}_{f}{\left(2N_{f}\right)}^{B}$$
where $\sigma_{a}$ is the reversing stress amplitude, ${\sigma^{\prime }}_{f}$ is the fatigue strength coefficient, $N_{f}$ is the number of cycles, and B is the Basquin exponent. Mechanical properties and material constants of the used material can be found in Table 1.

Estimates of total fatigue life determined by the Dirlik probability model for each type of PSDs are shown in Figure 9 and Figure 10. The plots show that if the window size is larger than 60%, the fatigue life increases exponentially for all cases. The accuracy of the estimate also decreases in areas where the overlap is less than 50% of the window size. It should be noted that the lowest value of the window size was 5% of all samples. However, testing has shown that it is not appropriate to reduce the window size further due to the loss of calculation accuracy. When taking a very small number of samples that enter the FFT calculation, the PSD values will be distorted and will not correspond to reality [36].

The presented results of the life calculations were performed for the Dirlik probability model. Similar behavior was found when using the Tovo–Benasciutti model. The difference in the extent of the damage using the Tovo–Benasciutti model and the Dirlik model are on average 0.35% to 8.43%, which can be considered acceptable. The differences between the models depending on the changing parameters of the Welch method are graphically shown in Figure 11. Note that when calculating the fatigue life, it is necessary to emphasize the correct choice of the probabilistic model. If the calculation used, for example, Rayleigh’s model, the results would be inaccurate, as this model is more suitable for narrowband signals. Therefore, the two most popular broadband load models were selected for fatigue analysis.

## 4. FEM Supported Fatigue Life Analysis

In order to verify the findings resulting from the sensitivity analysis described in the previous section, a FEM simulation of a simple component was performed. The analyzed object was a clamped beam of rectangular cross-section with two notches in the middle. A point mass of 50 g was applied at the free end of the beam, as shown in Figure 12. The beam was excited at the bond site. PSD type D from the previous analysis was used for the excitation. Since the lifetime estimation was performed in the frequency domain, i.e., using the spectral method, it is assumed that the excitation signal is stationary and ergodic, Gaussian-distributed and has zero mean value.

The lifetime calculation was performed in the FE-SAFE program. The input was data obtained from a preliminary simulation performed in Abaqus/CAE. The pre-simulation consisted of modal analysis and linear dynamic analysis. The output of the modal analysis was mode shapes, eigenfrequencies and modal stresses (Table 2). Three modes were identified in the range up to 500 Hz. The aim of the linear dynamic analysis was to obtain the generalized displacements and the phase angle of the generalized displacements for all modes (Figure 13). Note that unit excitation was considered in the calculation. From the response characteristics and modal stresses, real stress fields corresponding to unit excitation were obtained by modal superposition. If the actual excitation in the form of a PSD is subsequently included, it is possible to obtain stress PSDs, which enter into the fatigue life calculation. 

A total of 380 lifetime calculations were performed, in which the parameters of the Welch method were changed in accordance with the previous analysis. For this purpose, a computational loop was created in the Isight program (Figure 14). The power spectral densities entering the lifetime calculation were taken from the previous analysis in MATLAB. The outputs from the FEM pre-simulation were the same in each loop cycle. For this reason, the location of the damage did not change. However, the degree of damage varied depending on the setting of the Welch’s parameters. The location of the damage can be seen in Figure 15, which shows the result of the life calculation for the reference PSD. The reference value of the lifetime of the analyzed sample is approximately 43 s (10^1.631^ s). Figure 16 graphically shows how the lifetime changes depending on the setting of the Welch’s parameters compared to the reference value. It can be seen that the calculated lifetimes range from 11 to 603 s. The results show that the smallest errors in the estimate occur when the window size is less than 60% and the overlap is more than 50%. This is in line with previous findings.

## 5. Conclusions

The Welch method plays an important role in digital signal processing. It is commonly used to estimate the power spectral density. The Welch method reduces the variance of the periodogram method by averaging. From a practical point of view, the PSD is obtained by dividing the time series into overlapping subsegments, computing a modified periodogram for each subsegment and averaging the periodograms. Power spectral density is also an important characteristic in the analysis of fatigue life of components in the frequency domain. The aim of the paper was therefore to examine how the setting of Welch’s parameters affects the results of such an analysis. Three parameters were considered: type of weighting window, size of subsegment, and size of overlap of subsegments. To generalize the findings, calculations were performed for six different types of PSD functions. The results were verified on the example of FEM analysis of the service life of a notched beam.

Based on the obtained results, it can be stated that the best configuration of Welch method parameters for PSD calculation and the subsequent determination of lifetime by the spectral method is the smallest possible time window (maximum 60% of all samples) and the largest window overlap (minimum 70% of window size). The results show that the lifetime values in this area are smoothed and close to the reference value. At this point, it should be noted that the above findings are valid for the probability models used, i.e., Dirlik and Tovo–Bennasciutti.

These findings are particularly important in terms of the research and development of mechanical systems and components, whose fatigue life is crucial to ensure their long-term and safe operation. It has been shown that incorrect parameter settings can lead to significant errors in lifetime estimation in the frequency domain. Spectral methods are often used in engineering practice, as they provide a relatively good estimate and are not time consuming. For this reason, it is important to point out the need for the correct transformation of time data to spectral ones.

## Figures and Tables

**Figure 1 materials-16-01007-f001:**
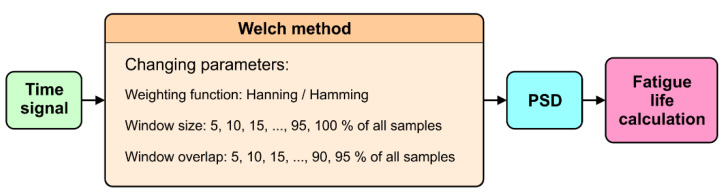
Design of experiment flow chart.

**Figure 2 materials-16-01007-f002:**
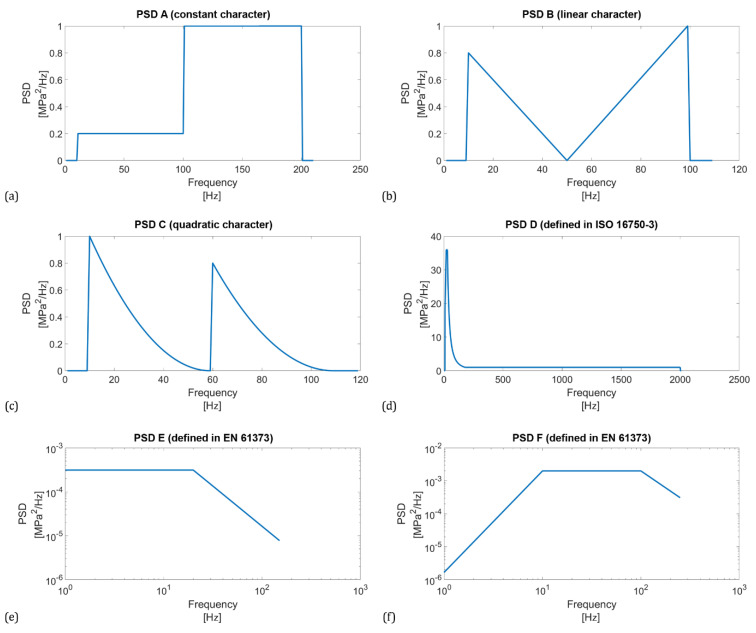
Synthesized PSD using constants (**a**), constants with linear connection (**b**), and constants with quadratic connection (**c**), standardized PSD functions according to ISO 16750-3 (**d**) and EN 61373 (**e**,**f**).

**Figure 3 materials-16-01007-f003:**
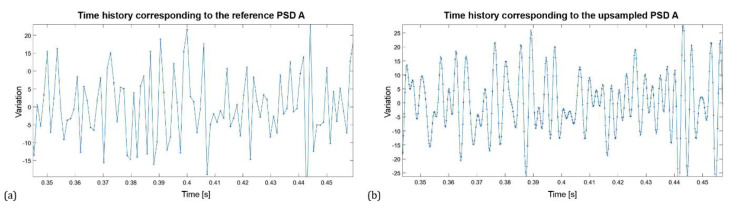
Time history obtained from the reference PSD A (**a**) and from the upsampled PSD A (**b**).

**Figure 4 materials-16-01007-f004:**
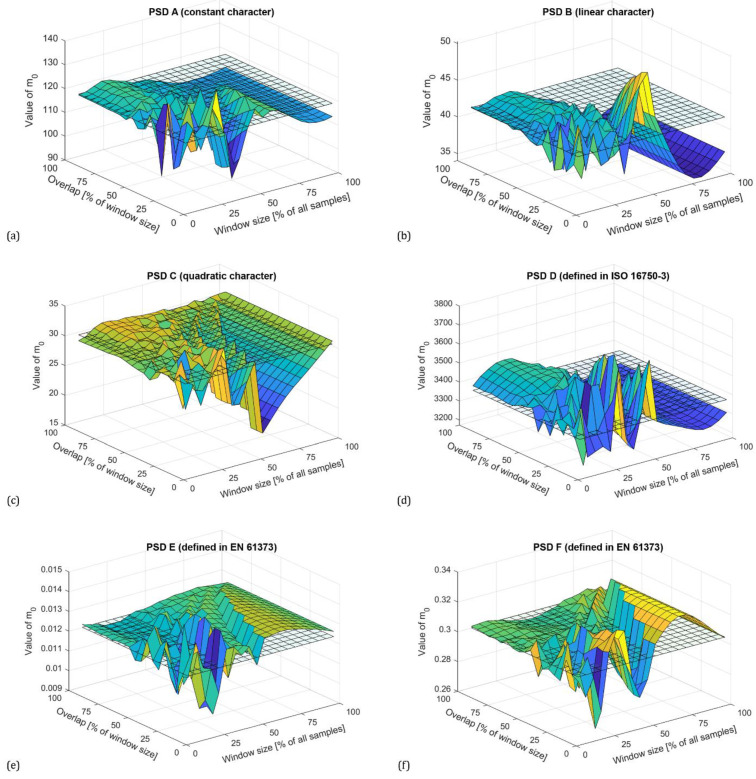
Zero spectral moments (variance) depending on window size and overlap for PSDs calculated with using Hamming window according to synthesized PSD using constants (**a**), constants with linear connection (**b**), and constants with quadratic connection (**c**), standardized PSD functions according to ISO 16750-3 (**d**) and EN 61373 (**e**,**f**).

**Figure 5 materials-16-01007-f005:**
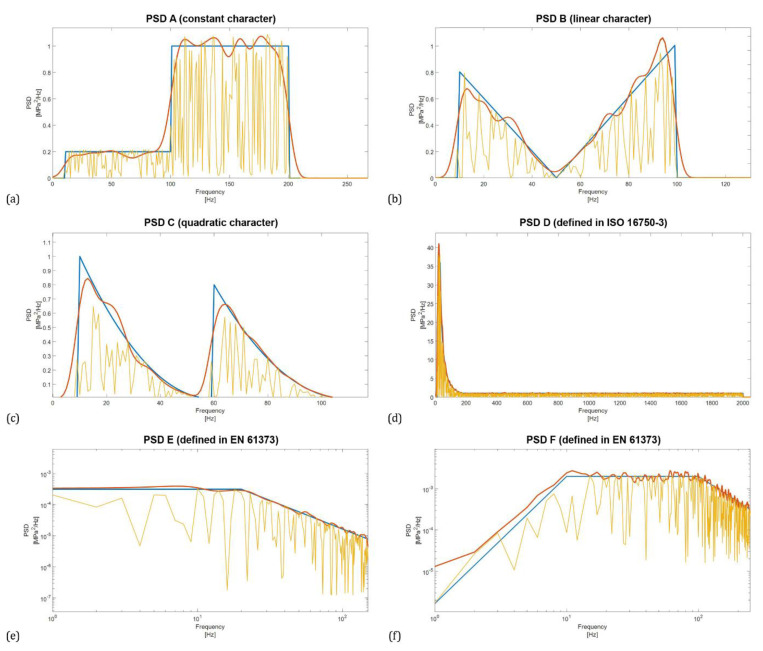
Comparison of PSDs calculated with different window size and overlay using Hamming window, blue line represents the reference PSD, red is for window size 20% and overlap 90%, yellow is for window size 85% and the overlap 25%. Comparisons prepared according to PSD using constants (**a**), constants with linear connection (**b**), constants with quadratic connection (**c**), standardized PSD functions according to ISO 16750-3 (**d**) and EN 61373 (**e**,**f**).

**Figure 6 materials-16-01007-f006:**
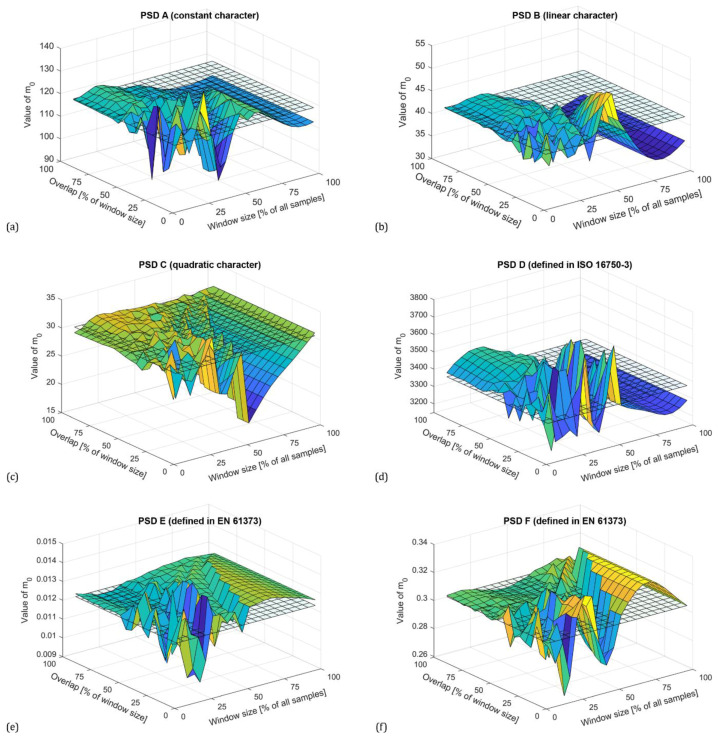
Zero spectral moments (variance) depending on window size and overlap for all PSDs computed using Hanning window according to synthesized PSD using constants (**a**), constants with linear connection (**b**), and constants with quadratic connection (**c**), standardized PSD functions according to ISO 16750-3 (**d**) and EN 61373 (**e**,**f**).

**Figure 7 materials-16-01007-f007:**
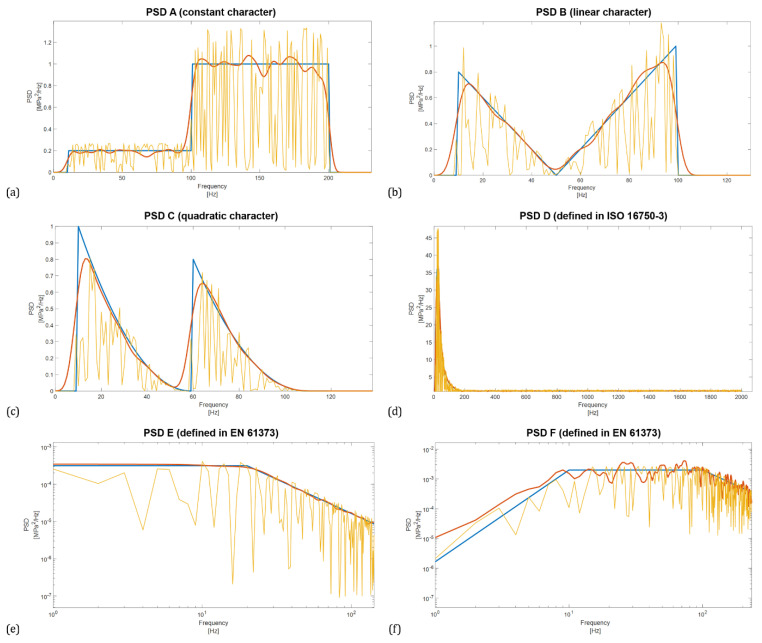
Comparison of PSDs calculated with different window sizes and overlay using Hanning window, blue line represents the reference PSD, red is for window size 20% and overlap 90%, yellow is for window size 85% and the overlap 25%. Comparisons prepared according to PSD using constants (**a**), constants with linear connection (**b**), constants with quadratic connection (**c**), standardized PSD functions according to ISO 16750-3 (**d**) and EN 61373 (**e**,**f**).

**Figure 8 materials-16-01007-f008:**
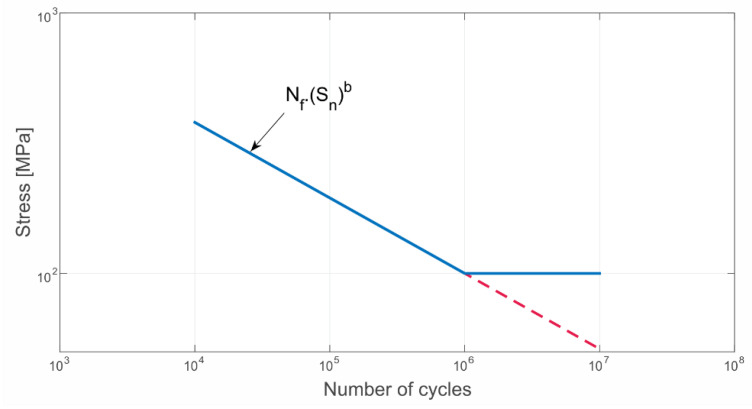
Idealized S-N curve defined using Basquin’s equation.

**Figure 9 materials-16-01007-f009:**
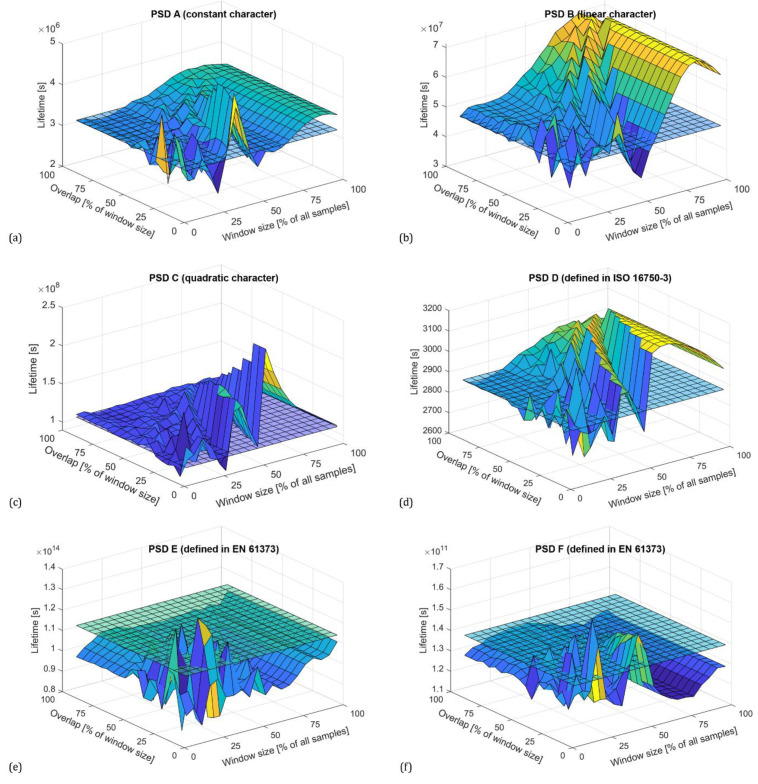
Fatigue life estimates determined by Dirlik probability model (Hamming window). Comparisons prepared according to PSD using constants (**a**), constants with linear connection (**b**), constants with quadratic connection (**c**), standardized PSD functions according to ISO 16750-3 (**d**) and EN 61373 (**e**,**f**).

**Figure 10 materials-16-01007-f010:**
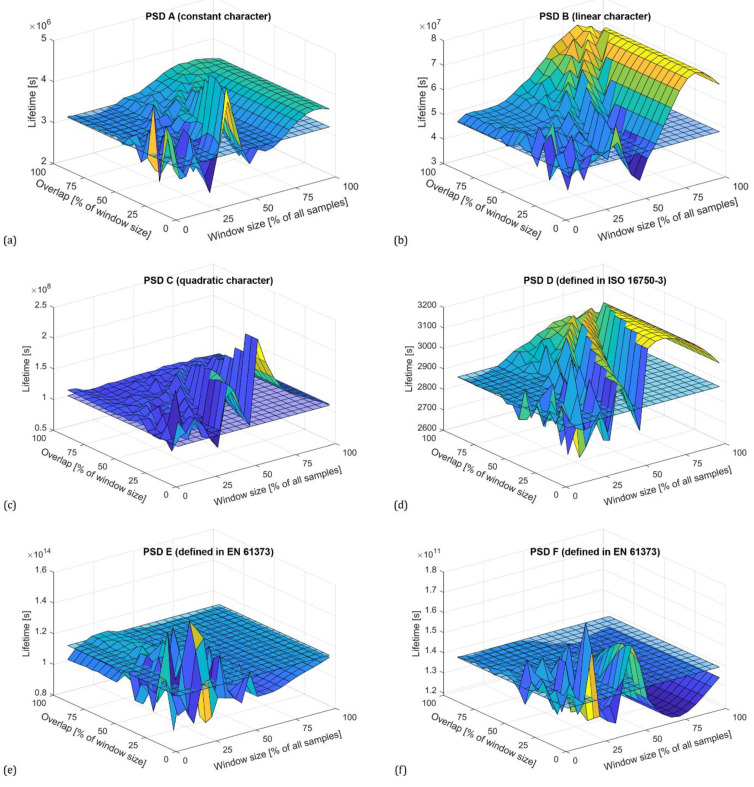
Fatigue life estimates determined by Dirlik probability model (Hanning window). Comparisons prepared according to PSD using constants (**a**), constants with linear connection (**b**), constants with quadratic connection (**c**), standardized PSD functions according to ISO 16750-3 (**d**) and EN 61373 (**e**,**f**).

**Figure 11 materials-16-01007-f011:**
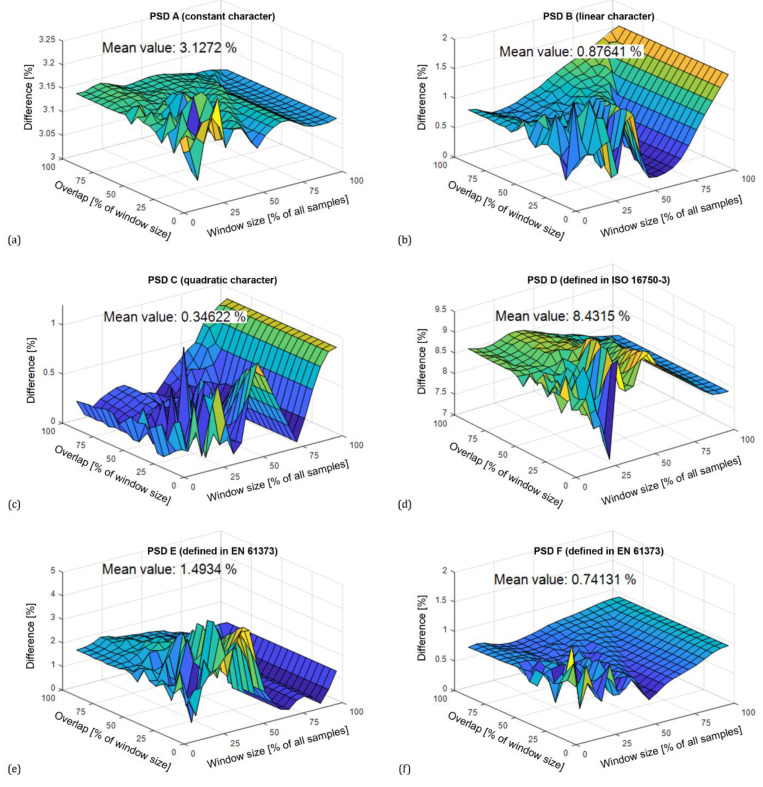
Percentage difference between Dirlik and Tovo–Bennasciutti models (Hanning window). Comparisons prepared according to PSD using constants (**a**), constants with linear connection (**b**), constants with quadratic connection (**c**), standardized PSD functions according to ISO 16750-3 (**d**) and EN 61373 (**e**,**f**).

**Figure 12 materials-16-01007-f012:**
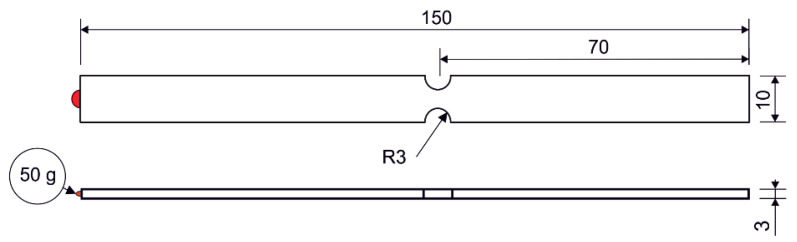
Specimen used in numerical simulation.

**Figure 13 materials-16-01007-f013:**
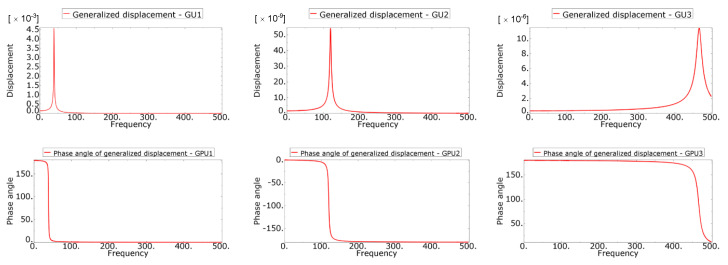
Linear dynamic responses.

**Figure 14 materials-16-01007-f014:**
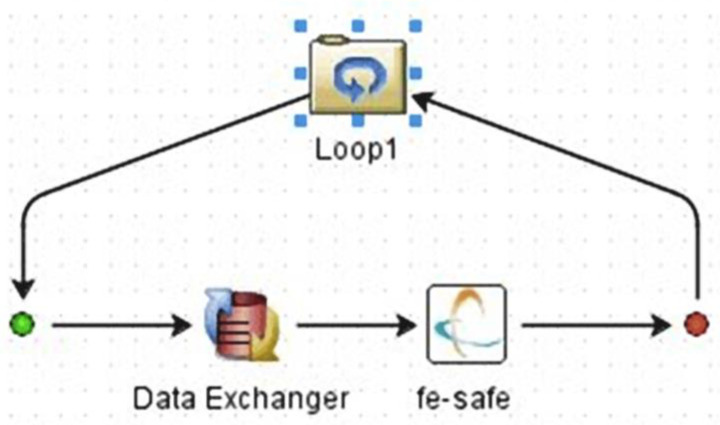
Isight computational loop.

**Figure 15 materials-16-01007-f015:**
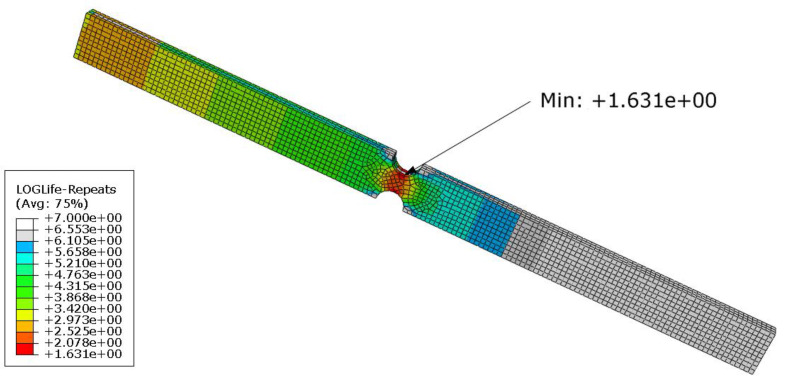
The results of fatigue life analysis for the reference PSD D.

**Figure 16 materials-16-01007-f016:**
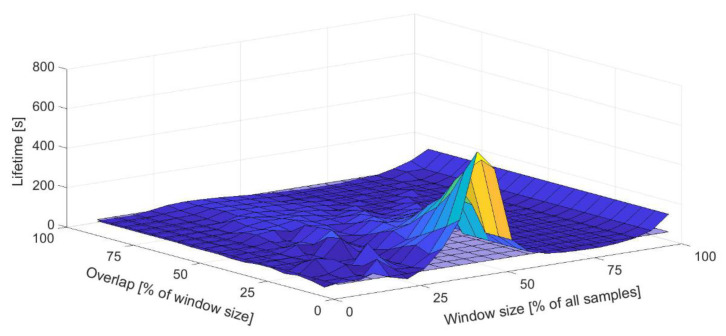
Matrix of lifetimes for different Welch’s parameters compared with the reference lifetime value.

**Table 1 materials-16-01007-t001:** Mechanical properties and material constants of the material.

Tensile Strength	Yield Strength	Elongation	E	ν	${\sigma^{\prime }}_{f}$	B
(MPa)	(MPa)	(%)	(MPa)	(-)	(MPa)	(-)
483	375	15	210,000	0.3	118	3.4496

**Table 2 materials-16-01007-t002:** Modal analysis results.

Mode	Mode Shape	Eigenfrequency [Hz]
1.	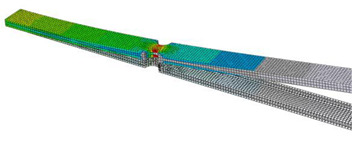	39.01
2.	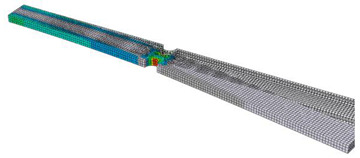	120.32
3.	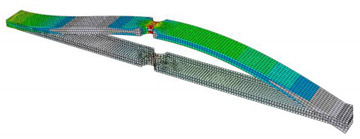	466.72

## Data Availability

Not applicable.

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
