# Peer review of "Influence of PSD Estimation Parameters on Fatigue Life Prediction in Spectral Method"

_materials, 2023, doi:10.3390/ma16031007_

Round 1
Reviewer 1 Report
The manuscript investigates the influence of power spectral estimation on fatigue life prediction. The work seems too simple to be published in a journal. All the presented theories are well known knowledge that can be found in the books or journals. Therefore, the innovation of the work is insufficient for a journal. The following comments can be considered.
-- The introduction is not well rewritten and not fits the requirement of a journal. Many references not very related to the work are excessively cited, such as L33-36, 130-104, 117-122. The authors should review the state the state-of-art of your topic and introduce the motivation of your work, not well-known basic concepts.
-- L42 what is the meaning of the ‘PSD is stable’?
-- Section 2 introduces related concepts.
-- Section 3 designs the experiment to check the influence of window size and overlap on fatigue life prediction. The reviewer thinks investigations of such parameters have little significance for vibration fatigue, but considering the frequency resolution in frequency domain or sampling rate in time domain makes more sense.
-- Section 4 provides a simulation example.
-- Section 5 presents the conclusion.
Overall, this work delivers very limited contribution to the vibration fatigue field. The quality of the manuscript is also not enough for publication. It may be more suitable as a conference paper.
Author Response
As Authors, we are grateful to the Reviewers for taking the time and effort to review our manuscript. We carefully read the comments on the text and took appropriate action. Changes have been made to the manuscript to improve it. We hope that the current version of the article meets the high standards of the journal Materials.
Below we present all the changes and answer the reviewers' questions.
Response to the reviewers
Reviewer #1: Comments and Suggestions for Authors
“-- The introduction is not well rewritten and not fits the requirement of a journal. Many references not very related to the work are excessively cited, such as L33-36, 130-104, 117-122. The authors should review the state the state-of-art of your topic and introduce the motivation of your work, not well-known basic concepts.”
Authors: Thank you very much for this comment. In the introduction, many articles on power spectral density estimation were cited, which was intended to familiarize readers with the current state of knowledge about PSD estimation. Of course, not all methods are used in mechanical calculations for fatigue life evaluation. We have reduced the number of cited papers and presented information. We also emphasize that the article deals with PSD estimation for future fatigue calculations, where the first 5 moments of the PSD function are used and play the most important role. This fact is very specific to fatigue calculations, and the impact of PSD estimation can be significant - greater than in other applications of this function.
The introduction was rearranged, redundant literature information was removed and information about the main purpose of the work was added.
“-- L42 what is the meaning of the ‘PSD is stable’?”
Authors: The word ‘stable’ refers to obtaining the same values of the power spectral density moments. We apologize for the insufficient explanation of this sentence. This has been corrected to the content of the article. Thanks for pointing out that sentence.
“-- Section 3 designs the experiment to check the influence of window size and overlap on fatigue life prediction. The reviewer thinks investigations of such parameters have little significance for vibration fatigue, but considering the frequency resolution in frequency domain or sampling rate in time domain makes more sense.”
Authors: Yes, it is true that all signal processing parameters affect the computational fatigue life. Also the sampling frequency and frequency resolution are very important. Unfortunately, usually we have no influence on these parameters if the signals are provided by third parties, which is often the case in cooperation with the industry. So we decided not to analyse these cases and focused on the parameters that we select when performing the simulation.
“-- Section 5 presents the conclusion. Overall, this work delivers very limited contribution to the vibration fatigue field. The quality of the manuscript is also not enough for publication. It may be more suitable as a conference paper.”
Authors: We agree with the reviewer that the subject matter presented in the work is very specific. However, it is an important and interesting topic for people dealing with signal processing for fatigue calculations. The authors did not find any other papers discussing the impact of PSD estimation parameters on fatigue design life. Because it is a specific topic, publication granted to an well-known open access journal gives the opportunity to reach the material by a people from around the world.
Reviewer 2 Report
Dear Authors,
Thank you very much for presenting this paper on power spectral density in fatigue analysis.
This is a highly specific topic with a relatively small readership.
Inclusion in the Materials journal entails some specifics - the article should be relevant to a wider audience and therefore some of my review will be guided in that direction.
Individual points:
The title of the article should not contain an acronym that is less well known - it is of course well known in our area, but not generally.
Related to this is the use of an abbreviation without explanation in the abstract - if you want to keep the abbreviation in the title you have to explain it in the abstract.
The abstract contains a great deal of general information, but it does not inform in any way about the result and the basic recommendation of the research.
Looking at the introduction and literature in general, I have to conclude that you have too many references that are duplicated because they contain the same information. This is not a literature review but an article and this number of references is unacceptable.
On the contrary, you lack some more application references and information about the reasons for your review.
For example:
10.1016/j.prostr.2018.12.314
10.3221/IGF-ESIS.49.10
There is not much to fault about the methodology and practical evaluation of the whole example.
The results are consistent with your assumptions and the chosen methodology.
In general your methodology is quite complex, although there is not much you can do to simplify it.
Most of the graphs require the addition of tables with extremes, which would help in evaluating the correctness of the conclusions.
Author Response
As Authors, we are grateful to the Reviewers for taking the time and effort to review our manuscript. We carefully read the comments on the text and took appropriate action. Changes have been made to the manuscript to improve it. We hope that the current version of the article meets the high standards of the journal Materials.
Below we present all the changes and answer the reviewers' questions.
Response to the reviewers
Reviewer #2: Comments and Suggestions for Authors
“The title of the article should not contain an acronym that is less well known - it is of course well known in our area, but not generally. Related to this is the use of an abbreviation without explanation in the abstract - if you want to keep the abbreviation in the title you have to explain it in the abstract.”
Authors: We agree with the reviewer, but want the title to remain in the proposed "compact form" and have clarified the abbreviation in the abstract. Thank you for the suggestion.
“The abstract contains a great deal of general information, but it does not inform in any way about the result and the basic recommendation of the research.”
Authors: In the abstract, we added information about the result and the basic recommendation coming from the study.
“Looking at the introduction and literature in general, I have to conclude that you have too many references that are duplicated because they contain the same information. This is not a literature review but an article and this number of references is unacceptable.
On the contrary, you lack some more application references and information about the reasons for your review. For example: 10.1016/j.prostr.2018.12.314, 10.3221/IGF-ESIS.49.10”
Authors: The number of references was limited and the necessary ones were selected. In general, also because of the suggestion of other Reviewers, the introduction was rearranged.
“Most of the graphs require the addition of tables with extremes, which would help in evaluating the correctness of the conclusions.”
Authors: The planned simulation experiment consisted of many computational steps and combinations, resulting in a large number of datasets. Therefore, we decided to show these results in the form of graphs showing only the trend observed when changing individual parameters. In some cases, a percentage difference is given (for Dirlik and Tovo-Benasciutti models). The numerical values would only refer to the given load case and could confuse readers who might think that such values would occur in every case. Please allow not to use numbers to suggest hard/numerical conclusions and more suggest qualitative rather than quantitative recommendations.
Reviewer 3 Report
The present paper deals with an analyses of the Welch method on the estimation of fatigue. The authors examine effects of parameters such as window size, weighting and overlap affects to the calculation of power spectral density. The paper is interesting and could be accepted for publication. Nevertheless, it needs a "Minor Revision".
1. It is given 94 References, it is better to reduce them by 50;
2. Compare the results with well-known ones;
3. Thee proposed method is not well justified;
Author Response
As Authors, we are grateful to the Reviewers for taking the time and effort to review our manuscript. We carefully read the comments on the text and took appropriate action. Changes have been made to the manuscript to improve it. We hope that the current version of the article meets the high standards of the journal Materials.
Below we present all the changes and answer the reviewers' questions.
Response to the reviewers
Reviewer #3: Comments and Suggestions for Authors
“1. It is given 94 References, it is better to reduce them by 50;”
Authors: The number of references was limited and only the necessary ones were selected. In general, also because of the suggestion of other Reviewers, the introduction was rearranged.
“2. Compare the results with well-known ones;”
Authors: In the literature one can find analyses concerning the influence of the window type used and other parameters of the Welch algorithm on the estimation of the power spectral density. The effect on smoothing and accuracy of PSD determination is usually discussed. However, it is difficult to find information on the impact of these parameters on material fatigue, which is determined on the basis of PSD. It is also the main driving force behind the creation of the publication.
“3. The proposed method is not well justified;”
Authors: The simulations carried out are directly related to the algorithm for determining the fatigue life using the spectral method, also known as vibration fatigue. Also, determination of durability based on FEM calculations is specific but well known in mechanical engineering. To be clear in our simulations the FE-SAFE and Abaqus/CAE programs were used and not scripts written by self in MATLAB, as in the point 3. Design of Experiment. To help readers understand the simulations performed, additional explanations are given and added in the body of the article.
Round 2
Reviewer 2 Report
I see that you have thoroughly revised the entire manuscript, and I also see that you have thoroughly revised the literature. But in my opinion you have not reflected all my recommendations from the first review and that is a pity.
Author Response
Dear Reviewer,
thank you again for your valuable comments and suggestions in line with our article. We regret that our fixes do not satisfy you in 100%. However, we had the difficult task of taking into account the recommendations of 3 independent reviewers, which were not always consistent. Nevertheless, the improvements you have suggested have been included as far as possible and will also guide us as we write future articles.
Yours faithfully,
Authors